# Models of Collaborative Governance: The City of Los Angeles' Foreclosure Registry Program

**Helen Morales \* and Jack Meek** 

College of Business and Public Management, University of La Verne, La Verne, CA 91750, USA;
jmeek@laverne.edu

\* Correspondence: helen.morales@laverne.edu

**Abstract:** The economic recession of 2007–2013 brought many challenges to nations and cities throughout the world. Los Angeles experienced a foreclosure crisis that brought instability in the real property market, resulting in property loss and loss of revenue from property taxes and increasing demands on city resources from blighted properties. The paper begins with a background of the problem related to blighted properties and proceeds a literature review related to the five phases to the development and implementation of a governance network. The paper then examines a case study—the City of Los Angeles Foreclosure Registry Program's governance network to reduce blight—to assess the phases taken to learn if the theory of network design offers meaningful direction and insight. The paper closes with an evaluation of the consistency regarding the literature related to the five phases of governance network development and its implementation by the City of Los Angeles.

**Keywords:** collaboration; governance networks; metropolitan administration; housing

## 1. Introduction

The economic recession of 2007–2013 brought many challenges to nations and cities throughout the world. One of the challenges was addressing home ownership and the almost immediate departure from residents who no longer could afford the housing they had purchased. For the City of Los Angeles, the 2008 foreclosure crisis brought with it not only instability in the real property market, resulting in property loss and loss of revenue by way of property taxes, but blighted abandoned properties increased demands on city resources as well as increased the incidents of criminal activity throughout communities.

On 8 July 2010, the City of Los Angeles enacted the Foreclosure Registry Ordinance (Ordinance Nos. 181185) as a mechanism to protect residential neighborhoods from blight resulting from the lack of adequate maintenance and security from properties in foreclosure, defined as the recording of a Notice of Default with the Los Angeles County Recorder's Office. The Ordinance was subsequently amended by Ordinance No. 183281 (Amended Ordinance) effective 20 December 2014. The Amended Ordinance, among other things, attempted to strengthen the ordinance requirements by providing for a proactive inspection requirement. The intent of the proactive inspection was to verify that foreclosed property, defined as Real Estate Owned Status, is free from graffiti, debris, rubbish, garbage, trash, overgrown vegetation or other similar material and is being maintained in a clean and sanitary condition, and otherwise free from blighted conditions.

In Los Angeles, the foreclosure crisis resulted in over 50,000 vacant and abandoned homes becoming magnets for blight and illicit activities that destabilized neighborhood and costs the City an estimated $1 billion (Fixla.org 2014). At the same time the City of Los Angeles was experiencing the foreclosure crisis, it was in the mist of budget crisis, resulting in staff reduction and imposed furloughs.

In spring of 2012, advocacy groups formed and spent time documenting blighted properties. Out of 4000 properties inspected, they documented 906 blighted properties, of which 271 were documented with severely blighted conditions. From an economic perspective, during this same time period, home owners lost an estimated $4.4 billion in home values and the City lost $27 million in property taxes (Fixla.org 2014).

## 2. Study Focus

The problem this study examines is to understand how local governments can respond to housing foreclosure issues that result from economic conditions. In particular, the study focus is on the characteristics of the Los Angeles response to the foreclosure challenge. The central research questions explored by this research are:

1. What patterns or phases of collective action describe the evolution of the Los Angeles response to the housing foreclosure crisis?
2. Do these patterns resemble known patterns of collective action, particularly governance network patterns?

The purpose of the study is to examine how the Los Angeles response exemplifies how metropolitan systems are developing collective action through networks to solve very complicated problems. Collective action among bureaucratic agencies calls upon the construction of network of stake-holders—in this case an array of city departments—to address the foreclosure problem from their particular missions and practices. The pattern developed from collective action is compared to the literature on network formation and implementation.

What follows is an overview of the methods employed in this study to identify the phases or patterns of network formation. Also discussed are methods employed in extracting evidence in regard to patterns. The review of the literature allows the establishment of expectations in regard to how governance networks are formed. These expectations are then compared to the case of the City of Los Angeles' governance network and implementation for the Foreclosure Registry Ordinance to reduce blight. A summary of findings from the case follows along with implications for governance network formation.

## 3. Methods

This study relies on two methodological approaches. The first is an integrative literature review that synthesizes existing literature on governance network formation and establishes the central findings in regard to the phases of governance network development (Snyder 2019; Davis et al. 2014; Grant and Booth 2009). This approach to the literature review allowed the author to establish expectations to be examined in regard to governance network formation.

The second methodology employed in this study is ethnographic. Ethnographers frame research "as a mode of discovery drawing questions from the field site itself" (Boellstorff et al. 2012, p. 32). According to Boellstorff et al. (2012) ethnographic studies that entail "years of in-depth observation and participation in the field, engage participants over time, from a variety of viewpoints and perspectives" (p. 37). One of the authors was a stakeholder in the formation of the governance network examined and was appointed to direct the foreclosure registry in 2012 and engaged with lenders and banking institutions on foreclosed properties related to multi-family rental properties for the City of Los Angeles. The authors worked to establish and manage the foreclosure registry governance network in an effort to reduce the number of blighted foreclosed properties and to reduce its associated affects to the surrounding communities.

### 3.1. Foreclosure Data

The authors utilized the foreclosure data obtained from RealQuest to verify all Notices of Default filed by banks and lending institutions in the City of Los Angeles. Other Core Logic data regarding

foreclosure starts were used to obtain statistical data on a quarterly basis to establish foreclosure trends. Data related to constituency reports and complaints of blighted properties was collected from advocacy groups, City Councilmember offices and the Mayor's office. Other foreclosure related data, including blighted properties, came from the following City Departments: The City Attorney's Office, Neighborhood Prosecutors Office, Department of Building and Safety, Public Works, Los Angeles Police Department, Los Angeles Fire Department, Department of Water and Power, the City Controller's Office, City Administrative Office and the City Legislative Analyst Office. Data was also obtained from the Foreclosure Registry which maintained records reported to the City by banks and lending institutions. Additional property related data was obtained from LUPAMS data base containing all real property data within Los Angeles maintained by the City. Policy directives were also provided by the Mayor, City Council, the Controller's Office and the General Manager of HCIDLA.

As the information was obtained from the various Departments and data sources, questions were drawn from the field itself related to the various theory on networks identified in the five phases. The researchers then identified the phases and utilized them to research the network to test whether the theory reflected the actual network.

The authors were able to investigate the system anchored in the culture and the complex arrangements in the day to day work including collaborating with all stakeholders directing the foreclosure registry program, collection of the data and establishment of policies and practices all related to the establishment of a governance network. The network consisted of complex array of large otherwise autonomous city departments. One of the researchers worked within the City culture of complex departments assessing the complex systems comprised of each City Department, the political environment and the network as a whole. According to Boellstorff et al. (2012), "ethnographers address systems of support anchored in culture," that organize investigation so that it can access complex systems (p. 38). Data was drawn from governmental sources from each of the Departments that included foreclosed and specific blighted conditions and policy directives from department managers and elected officials.

The goal of the data assessment and methodology was to offer some insight into the developmental patterns and sequences of governance network formation within a metropolitan system in response to a cross jurisdictional issue.

3.1.1. Theoretical Perspective: The Role of Networks Addressing "Wicked Problems?"

Moving from Government to Governance

Pubic administrators search for better ways to serve their communities. According to Dwight Waldo, "administration may be thought of as the major invention and devise by which civilized [people] in complex societies try to control their culture, by which they seek simultaneously to achieve-within the limitations of their wit and knowledge-the goals and stability and the goals of change" (as quoted in Meek 2009, p. 4). Meek (2009) argues that since governance can take different forms, and can be conceptualized in various ways to organize stakeholders and, therefore, government is no longer the central player (p. 8). Governance networks provide an opportunity to include the community in the conversation to address the goals and goal changes discussed by Dwight Waldo.

According to Newell and Meek (2005) "social problems have outpaced popular solutions." Public administrators work within what Frederickson called a "disconjunctive state" and as Putnam has called a "disassociated state" marked with a decline in social capital, collectively referred to as the disarticulated state by George Frederickson (Frederickson 1999, p. 702). This means a move away from government, and the old ways of doing something, to governance where public administrators work as facilitators, in conjunction with citizens and social organizations, to produce social goals and services. Paquet (2005) suggests that citizens, action groups, and the states can be catalysts to create *loose intermediation* of social capital required to create smarter communities (p. 139). According to

Keast et al. (2004) policy makers need to lay the foundation and step back out of the way and let the network process of collaboration to occur through flexibility and innovation (p. 367).

It is people within the networks that conjoin to get the work accomplished. Helco criticized the *iron triangles* theory, as missing the open *networks of people* (Helco 1978). Helco further finds that issue networks are comprised of a group of individuals with shared interests, with a common base of information and understanding of the issue, who then regard each other as knowledgeable that allow public policy to be refined, evidence debated, and alternative options worked out (pp. 102–4). Governance networks provide the ability to tap into the knowledge contained within the actors and to put it to work to allow citizens to invent ways to resolve their predicaments (Paquet 2005, p. 133) and solve "wicked problems".

Keast et al. (2004) find that one of the biggest challenges for governments is dealing with *wicked problems*, requiring a new way of thinking to resolve them (Keast et al. 2004, p. 363; Sørensen and Torfing 2011, p. 843). "Network structures will lead to fully integrated systems in which members see themselves as interdependent-working toward systemic change, and see that, although they represent individual organizations, their perspective is a holistic one" (Keast et al. 2004, p. 365). This holistic viewpoint is established through a common mission that is central to networks and, along with represented individual organizations and a unique structural arrangement, where participants are continually doing something creates a new way of thinking. This new way of thinking can better address the problems associated with wicked problems.

### 3.1.2. Collaborative Governance

Collaborative governance network's goal is to facilitate open communication between government and third-party service delivery partners to reduce fragmentation and address wicked problems (Mosley and Jarpe 2019). Wicked problems cross jurisdictions and government boundaries, governments are increasingly forming governance networks to work collectively to address their problems and reduce service delivery costs. Collaborative governance is an important tool that adds value to governments in addressing wicked problems. Collaborative governance seeks to promote democracy and make government more accountable by ensuring that all stakeholders are brought into the process. Koliba et al. (2011) refer to this process as democratic anchorage. Government networks can pose a threat to democracy if they are unplanned, unintended or ad-hoc manifestations of incremental action. Democratic anchorage ensures that the perspective of all stakeholder is included in the policy making process in order to ensure that the purpose for which the governance network was established is meet.

### 3.2. Phases in Building Networks

To respond to wicked problems, collective action is required. Collective action often called upon is cross-departmental, cross-agency, even cross-jurisdictional collective efforts. These efforts take the form of networks. Overtime, routines in these networks form bonds so as to act in a governing manner—at least in the form of communications and coordinated action. As these governance networks formed, scholars began to witness patterns of interactions in networks to informed network leaders on fundamental practices to ensure functioning among partners in the governance networks. In this paper, we refer to these patterns as phases.

The *first phase* is to design the network. Goldsmith and Eggers (2004) suggest that a network designer must "identify possible partners, bring all relevant stakeholders to the table, determine and communicate to all members the expectations of how the network will function, assemble and enmesh the pieces of the network, devise strategies to maintain the network, and activate it (p. 55)." They further provide that in designing a network one must start with determining the goals, tied to the mission, with a focus on the outcome-based public value they are attempting to create, the "destination not the path" (p. 56). They suggest the use of public officials with the capacity to convene parties, that would not normally connect, and the integration of technology which can bring actors together to create

more community services in order to step outside the box and design a space centered on outcomes. Networks must be designed to be flexible to allow them to adapt quickly to environmental changes.

In designing a network, administrators must select a mode of network governance. Provan and Kenis (2007) offer three modes of network governance: participant-governed networks, lead organizations, and network administrative organizations and proposes key predictors of effectiveness of network governance (pp. 234–36). The authors further propose that the selection of the three modes is dependent on the following four key structural and relational contingencies: trust, size (number of participants), goal consensus, and the nature of the task (specifically the need for the network level competencies) (p. 237). The authors further propose the following:

(1)　if the contingency factors are inconsistent with regards to the number of inconsistencies and the inconsistency with the governance form, the less likely that the form will be effective leading to either ineffectiveness, dissolution or change in form of governance;

(2)　shared governance will be most effective where trust is shared among network participants, high density trust, when there are relatively few participants (6 or less), when the network level goal consensus is high, and when the need for competencies is low;

(3)　Lead organizational network governance is most effective when trust is narrowly shared, low density, highly centralized trust, with moderate number of participants, when consensus is moderately low, and when need for competencies is moderate;

(4)　network administrative organizations are more effective when trust is moderately shared among the members, with moderate number of participants, and the level of competencies are high (Provan and Kenis 2007, p. 241).

The *second phase* is to identify and implement policy learning. According to Sabatier and Weible (2014), policy-oriented learning creates "Changes in the belief systems of coalition members that include not only the understanding of a problem and associated solutions but also the use of political strategies for achieving objectives" (p. 198). Learning is affected by the multiple attributes including the forums, the level of conflict between coalitions, stimuli and actors. The forum must possess openness in participation and members must share in a common analytical training and norms of conduct, or degree of professionalism. Cross coalition learning occurs best at the intermediate level of conflict, where "opposing coalitions are threatened just enough to attend to the issue and remain receptive enough to new information to increase the likelihood of cross coalition learning" (p. 199). The more stubborn or difficult an issue, the lower the expectation that cross collation learning will occur. The actors represent anyone who is attempting to influence its affairs. Actor's attributes can also play a significant role in the learning process. The more extreme an actor's beliefs, the less likely that they will be able to learn from their opponents. Policy brokers can play an important role to facilitate learning between opponents.

Individuals are bounded rationally and motivated by their belief systems. Weible (2006) finds that individuals have three tiered hierarchical belief system: (1) Deep Core Beliefs-Normative/fundamental beliefs that span multiple policy subsystems-very resistant to change; (2) Policy Core Beliefs-resistant to change but more pliable than Core Beliefs-perceptions and causes of subsystems problems, orientation on value priorities, effectiveness of policy instruments, proper distribution of authority between market and government; (3) Secondary Beliefs -empirical that relate to subcomponent of a policy subsystem-susceptible to change in response to new information and events (p. 99). It is these belief systems that are changed, often by external shocks that occur outside of a policy subsystem, over a long period of time, or through hurting stalemate-where participants view a status quo as unacceptable and run out of alternatives to achieve their goals, that ultimately result in the learning that occurs within networks (Weible 2006, p. 101). This is the new way of thinking and can aid in the wicked problems faced by administrators.

The *third phase* is to select the members of the network. Goldsmith and Eggers (2004) suggests that there needs to be an integrator who acts as a hub and is capable to reach across service lines, build

an intergovernmental network, and find internal management talent that can creatively configure the best possible solution. Governments must identify either an integrator internally, with qualifications necessary to meet the requirements, or look outside the agency, and weigh the risks associated with an outside provider, including that a contractor may go out of business or otherwise resign from the contract. The authors provide three models of integration: government as integrator—one who has the skill, knowledge and experience to successful create a network; prime contractor as integrator-provides the ability to reduce the upfront capital costs, to start up quickly and to have the flexibility for the unknowns; and third-party integrator—who is hired solely to manage the network. The authors caution that third parties add another layer of government between the funder and the client, it has risks in terms of stability and continuity of institutional knowledge (pp. 78–85).

Goldsmith and Eggers (2004) provide that members must have cultural compatibility-shared values, operational capacity-operational excellence or specialized expertise, and proximity to the customer- neighborhood ties that will provide access to the customer (pp. 64–69). Paquet (2005) suggests that in creating smarter communities' networks will tap into advocacy coalitions and groups that are already connected with the community (p. 133). Leaders in a network build trust and create new ways of working together with the power to get people to come to an agreement (Mandell and Keast 2009, pp. 173–75). Governance networks accomplish democratic anchorage by ensuring that the network includes different political constituencies, to assure full representation in our democratic society, and by ensuring that the network ascribes to a relevant set of democratic norms (Koliba et al. 2011, p. 240). Democratic anchorage provides legitimacy to the network and creates social capital.

The *fourth phase* deals with network management. McGuire (2002) offers four management behaviors to evaluate network management: activation, framing, mobilizing and synthesizing. Activating identifies the network partners and stakeholders. Framing is the process of facilitating agreements with the participants including the roles, rules and network values. Framing and activation occurs at both during the formation and when the network breaks down. Mobilizing behaviors are centered on creating commitment between the participants and external stakeholders and creating support for the network. Synthesizing behaviors are utilized by managers to create an environment that allows for the productive interaction among network partners. Synthesizing is often accompanied by reframing or when there is an attempt to change the perception of a network participant (McGuire 2002, pp. 602–4).

Agranoff (2003) offers ten lessons on how to manage networks: (1) Be a representative of your agency and the network; (2) take a share of the administrative burden; (3) operate by agenda orchestration; (4) recognize shared expertise-based authority; (5) stay within the decision bounds of your network; (6) accommodate and adjust while maintaining purpose; (7) be as creative as possible; (8) be patient and use interpersonal skills; (9) recruit constantly; and (10) emphasize incentives (pp. 28–31). Managing networks also requires big picture thinking, coaching, mediation, networking, risk analysis, contract management, ability to tackle unconventional problems, strategic thinking, interpersonal communications, project and business management and team building (Goldsmith and Eggers 2004, p. 158). Network managers must be able to think creatively, be highly adoptive to resolving problems, and create win-win situations (p. 165).

The *fifth phase* is ensuring innovation. Public administrators are called to be innovators. According to Sørensen and Torfing (2011), "innovation is always driven by social and political actors who are facing specific problems and demands and chose to exploit particular opportunities" (p. 844). It requires collaboration of the various actors and can be successful in breaking policy deadlocks and improving public service. Innovation is often driven by the size of government and its ability to absorb costs failure, increasing competitive pressures, strategic management, and a more rigorous measuring of outcomes (Sørensen and Torfing 2011, p. 845). Successful public innovation corresponds to the preferences of elected officials, how it makes life easier for public employees, and the degree to which it creates user satisfaction (p. 850).

The five phases outlined above offer a valuable guide for the development of governance networks. What follows is a case study on the implementation of this guide to the development of the City of Los Angeles' foreclosure registry network. First, the paper outlines the case examined that was embraced by the building of the network. This is followed by an examination of the phases that were established in building the network in relation to the phases outlined in the literature.

### 3.3. Research Informed Expectations on Building Networks

Based on the research outlined above (summarized in Table 1), the following expectations were developed in regard to the network formation of the Los Angeles Foreclosure response:

*Expectation One—The building of governance networks calls upon establishing trust within the network to obtain participation (Mandell and Keast 2009; Goldsmith and Eggers 2004);*

*Expectation Two—Central to building trust will be the role of a central leader (an integrator) to move policy and implementation forward (Goldsmith and Eggers 2004);*

*Expectation Three—New and expanding relationships among stakeholders need to be continually developed and constantly nurtured (McGuire 2002; Agranoff 2003);*

*Expectation Four—Stakeholder innovations within the network are important to overcome barriers found with bureaucratic structures (Sørensen and Torfing 2011).*

*Expectation Five—The formation of the network will be sequential, building on the strength of each phase of network formation (Goldsmith and Eggers 2004).*

**Table 1.** Phases in Building Networks.

| | Network Phases | Implications in Application | Comments |
|---|---|---|---|
| 1. | Network Design | Create a mission statement tied to goals focused on outcomes based on public value. Select appropriate network mode: participant governed; lead organization; or network administrative organization. | Establishes trust & ensures high level of goal consensus. |
| 2. | Policy Learning | Policy learning utilizing conflict to create learning | Creates effective policy |
| 3. | Selection of Members | Culturally compatible-shared values. | Creates trust, social capital & democratic anchorage. |
| 4. | Network Management | Framing, facilitating agreements, mobilizing & synthesizing. | Creates productive environments. |
| 5. | Ensuring Innovation | Utilize elected official's capacity to convene, makes work easier for PA. | Makes it easier to get the "buy in" and creates user satisfaction. |

References: Agranoff (2007); Goldsmith and Eggers (2004).

## 4. Results and Discussion

### 4.1. Assessing the Los Angeles Foreclosure Network Phases

What follows is an assessment of the network processes and events outlined in the case study and how these practices match with the phases of network development as outlined in the literature.

#### 4.1.1. Network Design

Blighted properties have a negative impact on communities with higher than normal crime rates, produce visual blight such as graffiti, litter, debris and debilitated properties that reduce

property values, creating a lower tax base, and impose a drain on city resources (Blankenship 2014, p. 1). Blight is applicable to the following typologies for policy domain for governance networks: community development and housing issues, health, crime, social welfare and government operations (Koliba et al. 2011, pp. 125–26). Blight is a "wicked problem" for the City of Los Angeles which City Council has identified as requiring a Foreclosure Registry Network. The following analysis of the City of Los Angeles' Foreclosure Registry network addresses the five phases emphasized in this paper.

The first phase dealt with the design of the network. Network can be defined in narrow terms as legally autonomous organizations that work together to achieve both their own goals and collective goals. In this respect, the Foreclosure Registry Ordinance was established "to promote the health, safety and welfare of the residents, workers, visitors and property owners of the City of Los Angeles, as well as protect the economic stability, viability and livability of neighborhoods in the City by requiring the registration and monitoring of defaulted and foreclosed residential properties" (Foreclosure Registry Ordinance 2010). The Ordinance further provides for a proactive inspection of foreclosed properties as it was believed that these properties had an increased potential to become blighted. Therefore, the Foreclosure Registry Ordinance was initiated to deal with foreclosed blighted residential properties after the wake of the Foreclosure Crisis and was placed under HCDILA for its implementation. Other City departments have similar goal objectives related to blighted properties and their effects such as the Los Angeles Police Department (LAPD), Los Angeles Fire Department (LAFD), Public Works-Department of Street Services, City Attorney-Neighborhood Prosecutors (CA-NP) and the City Attorney, Los Angeles Department of Building and Safety (LADBS), Los Angeles Department of Water and Power (LADWP). The Foreclosure Registry Network (FRN) therefore, became a network of City related departments and HCIDLA as their lead to deal with defaulted and foreclosed residential properties in the City of Los Angeles.

The FRN was a goal-directed network that was not a formal mandate, but a result of the City Council's objective to reduce blight through the establishment of the foreclosure registry ordinance. The City of Los Angeles' Controller, Ron Galperin's audit recommended City Departments to collaborate efforts to remedy blighted properties and their associated effects (Galperin 2014). The Controller further recommended a systems GeoRegistry that would display the data and allow for constituents to be made aware of foreclosed properties in their neighborhood. The FRN's design is a Lead-Organization governed network. HCIDLA is the lead by way of the City Council's establishment of the Foreclosure Registry Ordinance. Additionally, HCIDLA maintained the funds necessary to establish the network and to provide the staffing for the network through annual registration and fees from the foreclosing parties.

Prior to the establishment of the FRN, stakeholder Departments never collaborated. In fact, all of the stakeholder Departments worked independently with little or no interaction many of whom were not aware that the City collected data on foreclosed properties. HCIDLA utilized the City Councilmember Deputy to work as in integrator with the capacity to convene the parties. The GeoRegistry was the integration technology that was the basis for the FRN and the instrument that was established to convene the parties to collaborate on blighted and abandoned foreclosed properties.

Key predictors of effectiveness of network governance forms include trust, number of participants, goal consensus and need for network level competencies. For Lead organizations, trust can have low density because governance is highly centralized; number of participants is moderate, goal consensus is moderately low and the need for network level competencies is moderate. In the case of the FRN, trust is at a low density because the goals of each individualized City Departments are not tied to the FRN. Their work in dealing with blighted properties can and is often done independent of the FRN.

Efforts were made by HCIDLA to establish trust with other City Departments by identifying the needs of the other departments through stakeholder meetings to obtain goal consensus, through task force meetings, through trainings to connect the needs with the HCIDLA resources, and by being responsive to inquiries. HCIDLA was able to provide positive results when other City Departments contacted them requesting assistance with a foreclosed property that was a nuisance. By way of example, the City Attorney's Neighborhood Prosecutors Office contacted HCIDLA to assist in a

property that was illegal occupied by gang member squatters that were selling drugs and conducting prostitution on the property. There were three recent homicides at the property, which was located near an elementary school. HCIDLA was able to contact the lending institution that foreclosed on the property and collaborate their efforts with the City Attorney's Office to evict the squatters from the property in order to give possession to the lending institution. Once the bank obtained possession HCIDLA staff worked closely with the lending institution representatives to ensure that the property remained free of blighted conditions and that it did not pose a nuisance to the community. The lending institution immediately demolished the structures on the property, some of which had illegal construction, in order to maintain the property free from nuisance. The property was later sold and a new home was developed.

Through this event, and similar interactions between City Departments and HCIDLA foreclosure registry staff, trust was developed and nurtured. HCIDLA continued its interactions at various City Department task force meetings and participation in stakeholder meetings in order to establish trust. Trust continued through positive interaction and responsiveness of HCIDLA staff.

HCIDLA worked to establish trust with the City Departments by working to ensure compliance on all blighted properties. Their interaction and success increased the trust density and established a successful working relationship with other City Departments. This trust density continued to grow within the other City Departments. As a lead organization, HCIDLA established the network by identifying stakeholders and worked closely with them to identify concerns and operations on collective action. Regular, monthly, task force meetings were established, where the parties were able to interact and establish working relationships at the street bureaucrat level on an informal basis. The success of remedying blighted properties occurred most at this level. The number of participants for the FRN is moderate comprised of various City Departments that are affected by or deal with blighted residential properties.

### 4.1.2. Policy Development Strategy

The second phase relates to the identification and implementation of a policy development strategy. Networks require a framework that allows for continual leaning and the advocacy collation framework's main focus is policy-oriented learning. Learning within networks is affected by forums, level of conflict between coalitions, stimuli and actors. According to Sabatier and Weible (2014) in order to establish learning, the forum must possess openness, common analytical training, and norms of conduct or degree of professionalism. In the case of the FRN, most all participants where open and receptive to the FRN. The only member that was not was the Los Angeles Department of Building and Safety (LADBS). LADBS often did not attend the FRN meetings and attending only when they were prodded from the councilmember deputy.

Cross coalition learning occurs best at the intermediate level. HCIDLA and LADBS are two departments within the City that deal with housing issues and blighted properties. HCIDLA deals with multifamily units and LADBS deals with single family residents and vacant and abandoned residential properties. HCIDLA does not normally have jurisdiction over Single Family Dwellings (SFD). However, related to the FRP Ordinance, City Council placed the FRP Ordinance under the jurisdiction of HCIDLA only after LADBS had argued with City Council that they did not have the funding to operate the program citing the Cities budget crisis and staffing issues during the foreclosure crisis. This gave HCIDLA the jurisdiction over single family residential properties for the limited purpose of the FRP Ordinance. Additionally, historically tensions between the two departments were common as they perform similar actions however their jurisdiction is divided between Multi-family (HCIDLA) and SFD (LADBS). It should be noted that 90 percent of properties that are registered and subject to the FRP Ordinance, are SFDs, however HCIDLA, who did not previously maintain jurisdiction over SFDs, was charged with the implementation and management of the FRP Ordinance. Therefore, from its inception, tension existed between LADBS and HCIDLA prior to the creation of the FRP ordinance and FRN and these tensions were brought to the FRN. Sabatier and Weible (2014)

find that cross coalition learning can occur where conflict is present when the opposing coalitions are threatened to attend to the issue and remain receptive to new information. In the case of the FRN, LADBS was threatened enough to put into place a proactive inspection process, however collaboration never fully materialized because LADBS was not receptive to the new information and was only concerned with meeting the requirements of the ordinance. LADBS never fully became a member of the network in a cross-collaborative format to realize any gain from within or as a result of the interaction with the FRN. In the case with LADBS, deep core belief systems between it and HCIDLA made it very resistant to change.

### 4.1.3. Selection of Members

Phase three involves the selection of a network. A Councilmember and his Deputy staff worked as a political integrator to establish the members of the FRN along with HCIDLA management staff, as an integrator. City Council first identified City Departments when they established the Foreclosure Registry. Later the City Controller identified City Departments. The lists were combined and the Councilmember's Deputy Staff utilized her power to convene the parties for the first Stakeholder's meeting. The stakeholders collaborated and other City Departments were identified and the final stakeholder memberships was established.

The Councilmember's Deputy utilized her political power to convene the parties to attend stakeholder meetings and to otherwise collaborate with HCIDLA. Individual stakeholder meetings were held and HCIDLA worked directly with each City Department to understand each department need and identify opportunities for collaboration. HCIDLA was the lead organization that managed the stakeholder meetings and ultimately made the decisions. HCIDLA ensured democratic anchorage by ensuring that all Departments that dealt with blighted properties and the constituency was included. HCIDLA utilized advocacy coalition groups to engage the constituency, which mainly consisted of housing rights advocates and their members, who were Los Angeles constituents.

### 4.1.4. Network Management

The fourth phase dealt with network management. In order to evaluate the FRN's network management, it was helpful to utilize McGuier's 4 management behaviors: activation, framing, mobilizing and synthesizing. The FRN was activated by considering the effects of blighted properties and determining all City Departments that deal with blighted residential properties and the associated effects on the community. The City Councilmember's Deputy worked with HCIDLA in the framing process including the identification of stakeholders. The literature tells us that framing also is activated when the network breaks down. Often times through-out the stakeholder meeting process, there was a City Department or two that was reluctant or non-participatory which created a beak-down of the network. When this occurred the Council member's deputy utilized her political power to convene, and convened joint meetings where she required participation, and participation was obtained and mobilizing efforts ensued where commitments were made. Synthesizing behaviors were successfully employed in order to create an environment that would allow for continued collaboration. These behaviors included efforts made to change negative perceptions of code enforcement inspectors. HCIDLA code enforcement staff worked hard to establish working relationships with LAFD, senior LAPD lead officers and city attorney neighborhood prosecutors. HCIDLA code inspectors became members of the LAFD task force unit that worked closely with LAPD and the City Attorney's Office to collaborate efforts on blighted properties. HCIDLA management encouraged the continued participation with the TASK force meetings to allow the network to interact with other network participants. Management worked as the integrator in this process, but stepped out of the way and allowed the street level workers to find solutions. Reframing occurred with regular training programs for both LAPD senior lead officers and with the city attorney neighborhood prosecutors. HCIDLA management also established good communication and working relationships with all stakeholders.

Ten (10) network management lessons, the FRN did the following: (1) became a representative of both HCIDLA and the FRN; (2) took on all administrative burden; (3) operated by both agenda orchestration and individual communication and training that was not by agenda; (4) recognized shared expertise-based authority especially in the creation of the GeoRegistry; (5) since HCIDLA was the lead organization it was able to ensure that the FRN stayed within the decision bounds of its network; (6) HCIDLA continues to accommodate and adjust while maintaining its purpose, especially as it pertains to the continued collaboration of work on blighted properties; (7) attempts at creativity continued in day to day operations especially related to ensuring compliance of the FRO and blighted properties; (8) patience was wearing thin as the communication among other City Departments where often commenced with frustration over blighted properties especially from LAPD who dealt with associated criminal activity on many properties-HCIDLA staff continued to ensure that they did not overstep their bound even with the threat from other City Departments, especially as it related to eviction of tenants from blighted properties in the foreclosure process; (9) HCIDLA staff continued to recruit new members and add them to the FRN; (10) HCIDLA, as the Lead Organization continued to emphasize incentives with other City Departments. As an example, HCIDLA first worked to inform FRN participants of the ordinance requirements and then identify joint incentives on how the Foreclosure Registry ordinance could help them meet their objectives of reducing blight in City neighborhoods. One of the most effective tools was the use of incentives to get stakeholders to collaborate. In addition, HCIDLA enforcement of the penalty fee as a mechanism to obtain compliance on blighted properties also proved successful. The establishment of trust and relationships that were built over time with HCIDLA FRN staff, LAPD, LAFD and the City Attorney Neighborhood Prosecutors office produced collaboration allowing the participants to reduce blighted properties and its associated effects on the community.

### 4.1.5. Ensuring Innovation

Lastly, related to phase five, the foreclosure crisis provided the city with an opportunity to become innovative in its ways to address blight, especially in light of the city's budget crisis. The City of Los Angeles is large enough to absorb costs of failure, and more importantly many elected officials were a driving force of the FRN, including then Councilmen Eric Garcetti. The FRN brought together various City Departments to address blight in one forum. The foreclosure crisis created an increase in vacant and abandoned residential properties increasing incidents of blight and its associated crime. This placed a greater degree of stress on City street level workers, LAPD Senior Lead Officers, Pubic Works-Department of Sanitation Officers, LAFD, Neighborhood Prosecutors and others during a budget crisis, causing heightened tensions. Moreover, the constituents in communities throughout the City, were complaining that the blight was reducing their property values and increasing criminal activity in their neighborhoods. This heightened tension and frustration among affected city departments, created opportuning to choose and exploit other opportunities. The FRN was a choice which many city departments ultimately viewed as an opportunity that created an avenue to avoid deadlocks in operational daily tasks creating new ways of working effecting to create user satisfaction. The cross collaboration within the FRN established relationships and trust that created opportunities to collaborate, establish new work processes, and successfully address blight. The stakeholders were able to seize upon the relationships established by HCIDLA with lenders, beneficiary, trustees and their representatives. It is important to note that the GeoRegistry, a systems program that was established to create stakeholder collaboration did not contribute to collaboration. Many of the stakeholder never used the GeoRegistry system, however the relationships that were established through the FRN continue to grow and reframe themselves into further opportunities for stakeholders to collaborate.

### 4.2. Assessment of Network Phases in Building the LA Governance Network

HCIDLA worked to create a governance network that brought City Departments together with the goal of collaborating to address blighted properties in the foreclosure process. Each of the stakeholder

departments had their own goals dealing with various issues either directly related to blight or effects resulting from blight. In relation to phase one, Network Design, the FRN was created, as legally autonomous organizations that work together to achieve both their own goals and collective goals. All City Departments worked to remedy blight and its associated crime. For example, the City Attorney Neighborhood Prosecutors office prosecuted owners of blighted properties while the LAPD responded to calls resulting from criminal activity associated with blighted properties, HCIDLA collected the data of ownership information and monitored the property condition and worked with lenders and defaulting properties to ensure that properties remain free from blight. However, collectively with the FRN, the City Departments collaborated to resolve blighted properties by utilizing the contact information obtained through the FRO, the relationships established by the FRP staff and owners of foreclosed properties and those in the foreclosure process, and the other City Departments.

Network functioning, which is how network conditions lead to various network level outcomes, is important in order to understand why networks produce certain outcomes. The authors define network effectiveness as the positive network outcomes that could not have been achieved independently, without the collaboration. In the case FRN, HCIDLA maintained a database of registration information listing direct contact lender information for properties in the foreclosure process that was not available to other City Departments. HCIDLA did not have access to all the nuisance properties in the foreclosure process. However, other City Departments and their street level bureaucrats, had direct contact with blighted properties and their associated criminal activity. Neither HCIDLA nor the other City Departments were able to independently be effective, however their actions in collaboration effectively produced results in remedying blighted properties in the foreclosure process.

What can most be gleaned in the administration of the selected five phases of governance addressed in this paper are related to the powers to convene and the implementation of the GeoRegistry systems vs the actual FRN established in the process. Table 2 reflects the implementation of the five phases and provides in the comments suggestion some findings resulting from their implementation. The role of management is critical for an effective network governance especially as it relates to tensions within the network. Tensions within the FRN were resolved most effectively by the political integrator who was able to convene the City Departments to participate. When the Councilmember integrator, who held the power to convene the City Department, was removed from the FRN, the City Departments that held deep core values, regardless of the benefits that were identified for the City Department to participate, withdrew from the FRN. Therefore, in analyzing the case of the City of Los Angeles and the FRN, we find that the integrator with the power to convene the parties was a critical step in the success of the FRN. The trust and relationships that were established at the street level produced results through the continued interaction of the stakeholders. The GeoRegistry system had no effect on the FRN because the stakeholder did not utilize the program. The interaction between stakeholders and the relationship building through task force meetings and trust that subsequently developed, produced results that reduced blight in the community.

**Table 2.** Phases of Network Building and Case Illustration.

| Network Phase | LA Case | Comments |
|---|---|---|
| 1. Network Design | Each of the stakeholder departments were legally autonomous organizations that worked together to achieve both their own goals and collective goals. HCDLA worked as the Lead Organization within a goal directed network to remedy blighted and abandoned residential property in the wake of the 2008 foreclosure crisis. The FRN trust was low density as HCDLA held a highly centralized governance and other Stakeholder's performance was not tied to the FRN. | HCIDLA was required to establish trust within the network to obtain participation within the network. Some City Departments expressed reluctance due to their experience in working with other City departments that proved to be ineffective and often times non-responsive. This required continual reminders of incentives for collaboration as well as establishing trust by creating working relationships with street level bureaucrats that produced results. |

**Table 2.** *Cont.*

| | Network Phase | LA Case | Comments |
|---|---|---|---|
| 2. | Policy Learning | HCIDLA implemented a learning policy strategy in order to establish cross coalition learning due to the established Los Angeles City bureaucratic structure that limited collaboration among and across City Departments. | Conflict helped to create learning opportunities in a bureaucratic structure that limits interaction among City Departments. However, without the integrator with political backing, the relationship among the two departments with the most bureaucratic structures, due to the deep core beliefs, could not be penetrated. |
| 3. | Selection of the Members | The FRN was established by a political mandate, City Ordinance and later by the Controller Audit and established by a Councilmember integrator. The FRN was composed of culturally compatible City Departments with shared values related to dealing with blighted properties. The City ensured democratic anchorage by ensuring that all affected City Departments as well as the constituency was included in the process. HCIDLA utilized an established advocacy coalition group consisting of housing rights advocates to represent the constituency. | Trust was created within the network as a result of the shared values of each department to address blighted properties. The integrator assisted in moving the FRN forward by ensuring that the members came to the table to listen and participate. |
| 4. | Network Management | FRN was activated by the identification of all City Departments affected by blighted and abandoned properties. The FRN was framed by the Chair of the Housing Committee, a City Councilmember Deputy, both in getting the stakeholders to the table and when the FRN was breaking down and mobilizing stakeholders. Synthesizing behaviors were established with Code Inspectors, Senior LAPD lead officers, and LAFD personnel. HCIDLA became a representative; took on all administrative burden; operate by both agenda and individual communication; recognized shared expertise; ensured the FRN stayed within decision bounds; continued to accommodate and adjust; ensure creativity; establish patience; continued to recruit new members; and emphasize incentives. | Productive environments were created at the street level bureaucrat level establishing working relationships that were non-existent prior to the creation of the FRN that produced results thereby incentivizing participation. |
| 5. | Ensuring Innovation | The FRN capitalized on the fact that it had funds to establish a GeoRegistry system as a mechanism to address blight. The establishment of the FRN was created as a means to establish the GeoRegistry, however the collaboration and success of dealing with blighted and abandoned properties occurred within the FRN. The opportunities to establish trust and create networks of street level bureaucrats to effectively reduce the effects of blighted and abandoned foreclosed properties were established through the exchange of shared values. Opportunities where established through operational daily tasks that seized on relationships already established by HCIDLA with lenders, beneficiaries, trustees and their representatives. | Innovations were established that broke down the bureaucratic structures and allowed for cross collaboration through the FRN. The GeoRegistry that was the focus of the FRN, however has not proven as useful as the relationships that were created. |

## 5. Summary of Findings

It is clear that the LA Foreclosure program was addressing a wicked problem that called upon various departments of the City of Los Angeles to address. It is now recognized in the literature that these kinds of collective action are becoming common place. Based on the five phases of network development and the application of the case study, we can now turn to an evaluation of the consistency (or dissidence) that can be observed from the five-phase framework of network development as compared to the implementation of network governance by the City of Los Angeles with the development of its Foreclosure Program.

In addressing research question one—what patterns or phases of collective action describe the evolution of the Los Angeles response to the housing foreclosure crisis—this paper drew upon the case

study to identify five phases of development. In regard to addressing research question two—do these patterns resemble known patterns of collective action, particularly governance network patterns—the paper outlines four expectations based on the literature that we can expect in regard to network formation. Below is a list of study findings in regard to each expectation.

*5.1. Expectation One: The Building of Governance Networks Calls Upon Establishing Trust within the Network to Obtain Participation*

While common place, the building of governance networks is not easy. There are numerous barriers that must be addressed in the building of networks. As we observed in the LA Foreclosure example, HCIDLA was required to establish trust within the network to obtain participation within the network. Some City Departments expressed reluctance due to their experience in working with other City departments that proved to be ineffective and often times non-responsive. This required continual reminders of incentives for collaboration as well as establishing trust by creating working relationships with street level bureaucrats that produced results.

*5.2. Expectation Two—Central to Building Trust Will Be the Role of a Central Leader (an Integrator) to Move Policy and Implementation Forward*

Building trust among stakeholders in the network is central to building network competence. In the LA Foreclosure case, trust was created within the network as a result of the shared values and goals of each department to address blighted properties. The integrator assisted in moving the FRN forward by ensuring that the members came to the table to listen and participate. The building of trust among stakeholders is consistent with theories of governance network formation and practices. However, the role of the network integrator with the power to convene the parties was a critical step in the success of the Foreclosure Registry Network.

*5.3. Expectation Three—New and Expanding Relationships Among Stakeholders Need to Be Continually Developed and Constantly Nurtured*

New and expanding relationships among stakeholders need to be continually developed and constantly nurtured. In the LA Foreclosure case, productive environments were created at the street level bureaucrat level establishing working relationships that were non-existent prior to the creation of the FRN that produced results thereby incentivizing participation.

Despite constant attention to network development, there are limits to cross departmental collaboration. In the LA Foreclosure case, bureaucratic structure continually placed limits interaction among City Departments. Even with political integrator, the relationship among departments with the most bureaucratic structures, due to the deep core beliefs, could not be penetrated.

*5.4. Expectation Four—Stakeholder Innovations Within the Network are Important to Overcome Barriers Found with Bureaucratic Structures*

Innovations are important to overcome barriers found with bureaucratic structures. In the LA Foreclosure case, innovations where established that broke down the bureaucratic structures and allowed for cross collaboration through the FRN. The GeoRegistry that was the focus of the FRN, however has not proven as useful as the relationships that were created. However, the participation in task force meetings and establishment of trust within the FRN proved to be an innovative idea among City Departments that historically never collaborated before the creation of the FRN.

*5.5. Expectation Five—The Formation of the Network Will Be Sequential, Building on the Strength of Each phase of Network Formation*

In the building of governance networks, important challenges can be found in each of the phases that practitioners need to attend. These can be overcome, but will call upon diligent leadership efforts and a constant attention to the building of stakeholder inclusion and trust. Based on this research,

the central finding from this study is that it is clear that network formation is an iterative process—not a standardized sequential process—one that requires constant attention and management to ensure inclusion and participation. While phases of network development may occur, they are also revisited in order to maintain network sustainability.

**Author Contributions:** The authors seek to contribute to informing collective action in regard to addressing challenging issues facing metropolitan arenas. Both are committed to exploring the benefits and challenges of collaborative public management. Both authors contributed to this article and its conceptualization: methodology; software; validation; formal analysis and investigation, to the writing and preparation of the original draft.; writing—review and editing; visualization; supervision, and project administration.

**Funding:** This research received no external funding.

**Conflicts of Interest:** The authors declare no conflict of interest.

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
