# Peer review of "Models of Collaborative Governance: The City of Los Angeles’ Foreclosure Registry Program"

_admsci, doi:10.3390/admsci9040083_

Round 1

Reviewer 1 Report

I found the topic to be interesting, but expected a bit more regarding collaborative governance.  The "Theoretical Perspective" section seems disjointed in a major way.  There's no clear cohesion between what's discussed in this section and the case study/findings.  It's as though the author took a kitchen-sink approach and put as much information in the section as possible without consideration of how it all works together.  There's little commentary on the importance or value of governance networks.  The author(s) note that the networks provide a way to include the community, but doesn't explain why this is important.

The section "Phases in Building Networks" seems relevant and sufficient as related to the conclusions.

As for the case study, it was difficult to understand without the requisite background.  More information about the registry itself and how it came to be provides important context.  Some of this information is written, but buried in results which is curious.  The question I have is why the author provides results and discussion before explaining the methodology.

As for the methodology, it's unclear as to how the methods relate to the findings.  For example, the author(s) notes, "Data was drawn from governmental sources".  What data?  What governmental sources?  How is the ethnographic framework applied to the case?

Author Response

Thank you for taking the time to review and comment on this paper.  Your comments assisted to better guide and direct this paper in a more productive manner.  All of your comments have been taken into consideration and modifications made. The following details the specific modifications made in response to your comments:

A bit more regarding collaborative governance. 

Response: At Page 3 the following was added to address collaborative governance:

Collaborative governance network’s goal is to facilitate open communication between government and third-party service delivery partners to reduce fragmentation and address wicked problems (Mosley and Jarpe, 2019).  Wicked problems cross jurisdictions and government boundaries, governments are increasingly forming governance networks to work collectively to address their problems and reduce service delivery costs. Collaborative governance is an important tool that adds value to governments in addressing wicked problems. Collaborative governance seeks to promote democracy and make government more accountable by ensuring that all stakeholders are brought into the process. Koliba, Meek and Zia (2011) refer to this process as democratic anchorage. Government networks can pose a threat to democracy if they are unplanned, unintended or ad-hoc manifestations of incremental action. Democratic anchorage ensures that the perspective of all stakeholder is included in the policy making process in order to ensure that the purpose for which the governance network was established is meet.

The "Theoretical Perspective" section seems disjointed in a major way.  There's no clear cohesion between what's discussed in this section and the case study/findings.  It's as though the author took a kitchen-sink approach and put as much information in the section as possible without consideration of how it all works together. 

Response:  The Theoretical Perspective does appear to be kitchen-sink.  Therefore, in response to your point, the authors modified the sections to break down the Theory Discussion with a focus on 1. Moving from Government to Governance; 2. Collaborative Governance; and 3. Phases in building networks.

There's little commentary on the importance or value of governance networks.  The author(s) note that the networks provide a way to include the community, but doesn't explain why this is important.

Response:  this was included in the paragraph added at page 3, referenced hereinabove at number 1.

As for the case study, it was difficult to understand without the requisite background.  More information about the registry itself and how it came to be provides important context.  Some of this information is written, but buried in results which is curious.

Response: The following paragraph was added to include additional background information:

In Los Angeles, the foreclosure crisis resulted in over 50,000 vacant and abandoned homes becoming magnets for blight and illicit activities that destabilized neighborhood and costs the City an estimated $1billion (Fixla.org, 2014).  At the same time the City of Los Angeles was experiencing the foreclosure crisis, it was in the mist of budget crisis, resulting in staff reduction and imposed furloughs. In spring of 2012, advocacy groups formed and spent time documenting blighted properties. Out of 4,000 properties inspected, they documented906 blighted properties, of which 271 were documented with severely blighted conditions. From an economic perspective, during this same time period, home owners lost an estimated$4.4 billion in home values and $27 million in property taxes (FixLA.org, 2014).

The question I have is why the author provides results and discussion before explaining the methodology.

Response: The Methodology section was moved after the Theoretical Prospective and before the results. 

As for the methodology, it's unclear as to how the methods relate to the findings.  For example, the author(s) notes, "Data was drawn from governmental sources".  What data?  What governmental sources?  How is the ethnographic framework applied to the case?

Response:  The methods section was modified to include all government data obtained and utilized in the research. The method section was modified to add a new approach: an integrative literature review that synthesizes literature on governance network formation and establishes the central findings in regard to the phases of governance network development. A new section entitled Foreclosure Data that details the data obtained from government sources was also added.

Reviewer 2 Report

Review of Models of Collaborative Governance: The City of Los Angeles’ Foreclosure Registry Program

This paper analyzes the case of City of Los Angeles’ foreclosure registry program using theories of network. It especially focuses on how theories of network can provide meaningful insight in understanding collaborative public management. The way the City of Los Angeles initiated and managed the foreclosure registry program involves inter-departmental collaboration and the authors examines this aspect of collaborative public management through ethnographic research methods.

The case itself is sufficiently interesting both to network scholars and collaborative public management scholars as well as to general public administration scholars. However, the manuscript in its current form is not publishable and the authors need to polish the paper better. Below, I share my comments regarding how to strengthen the paper.

Major points

Theory

1) Phase 1-3 flow naturally, but 4 and 5 seem to come out of context. Regarding phase 4, why does policy development framework come after selection of members and network management? Shouldn’t this framework set up early in the network formation process (e.g., goal set up)? Learning and interaction can certainly play out after/during network management, but I am not sure why you call this “policy development framework.” Perhaps, change this to something akin to “policy learning.” Realtedly, I would not incorporate ACF unless there is a reason you have to. It is very odds that the authors suddenly cite Sabatier & Weible (2014) and focus on policy-oriented learning.

Regarding phase 5, why should the network governance result in innovation? It seems that the authors tried to lay out theoretical framework based on the empirical observations on how the LA’s foreclosure registry program management progressed. As such, the components of theory section are convoluted. They are not well connected to each other.

2) The literature review is unnecessarily long (many parts are irrelevant to the focus of this study) and it doesn’t generate an interesting research question. The authors should consider streamline this section and be succinct in proposing the five phases.

3) The five expectations spelled out in page 7 are not consistent with the five phases laid out earlier. I am not sure where these expectations were generated. This is one of the weakest part in the paper.

Analysis

4) This part should focus on the analysis. It should not introduce new concepts or arguments/findings from previous work. In many places, the authors do this, and hence it gives an impression that this section is mingled with the literature review. (e.g., discussions on Provan and Kenis (2007) in page 8 and Agranoff (2013)’s 10 network management lessons in page 10). I would just focus on the analysis based on the hypotheses proposed from the previous section.

Methods

5) This section should come first, not after the analysis. Currently, the authors use this section to justify the choice of ethnographic methodology. But what is missing is how the authors employed this ethnographic method. Plus, the authors’ justification of this methodology is far short of expectations. How do I know what the authors describe (regarding the program process) is correct? The authors argue that their descriptions are a result of years of in-depth observation and participation in the field (page 15). If so, the authors should provide a full description about how they got involved in observing (and participating in? what does this mean?) the program. The choice of ethnographic methods per se does not justify whether the authors’ observations and/or interpretation is valid. There should be some cross-references that can validate the authors’ findings. Or, at the very least, the authors are burdened to provide materials regarding how they tried to validate their interpretations. For example, “Data was drawn from governmental sources and policy directives” (p. 16). What data are you talking about? What governmental sources and policy directives were used? Also, what analytic approach the authors specifically (or strategically) used should be explained to augment the strength of ethnographic methods.

In sum, a more thorough method section that enhances credibility of the data sources and analysis needs to be provided rather a couple of self-justifying claims at the end.

Minor points

Background section look quite thin. I at least expected a couple of more sentences or one/two more paragraph about the background. It is very odd to see the next paragraph suddenly introduces the setup of the paper without the background having been sufficiently provided. The link to wicked problem is not well incorporated in the body. It was just used as a background. I would either delete it or incorporate this discussion somewhere in the analysis/discussion if the authors want to keep this part. “Cross-departmental collaboration” seems to be a better word than “network,” as this is essentially what the authors are examining. The term “phase” sounds awkward. Name it. Numerous grammatical errors and broken sentences degrade the quality of this manuscript. The authors should proofread the next iteration of this manuscript. For example, not “relative to”, but “related to.”

Author Response

Thank you for taking the time to review and comment on this paper.  Your comments assisted to better guide and direct this paper in a more productive manner.  All of your comments have been taken into consideration and modifications made. The following details the specific modifications made in response to your comments:

Theory

Phase 1-3 flow naturally, but 4 and 5 seem to come out of context. Regarding phase 4, why does policy development framework come after selection of members and network management? Shouldn’t this framework set up early in the network formation process (e.g., goal set up)?

Response: This is a great point.  In response, phase 4 was moved to phase 2 and the remainder of the phases were moved down in sequence.

Learning and interaction can certainly play out after/during network management, but I am not sure why you call this “policy development framework.” Perhaps, change this to something akin to “policy learning.” Realtedly, I would not incorporate ACF unless there is a reason you have to. It is very odds that the authors suddenly cite Sabatier & Weible (2014) and focus on policy-oriented learning.

Response:  This point is well taken.  This section was renamed policy learning and the manuscript was modified to the focus of policy-oriented learning.

Regarding phase 5, why should the network governance result in innovation? It seems that the authors tried to lay out theoretical framework based on the empirical observations on how the LA’s foreclosure registry program management progressed. As such, the components of theory section are convoluted. They are not well connected to each other.

Response:  Innovation is relative to a current discussion on collaborative governance in a time and age where the current tools available for public administration do not meet the current needs. Innovations can also work as incentives to encourage greater collaboration.   

The literature review is unnecessarily long (many parts are irrelevant to the focus of this study) and it doesn’t generate an interesting research question. The authors should consider streamline this section and be succinct in proposing the five phases.

Response:  The Theoretical Perspective does appear to be kitchen-sink.  Therefore, in response to your point, we modified the sections to break down the Theory Discussion with a focus on 1. Moving from Government to Governance; 2. Collaborative Governance; and 3. Phases in building networks. In addition, a new section in the introduction was added entitled study focus in order to provide more focus to the paper.  In that section two research questions were added. 

The five expectations spelled out in page 7 are not consistent with the five phases laid out earlier. I am not sure where these expectations were generated. This is one of the weakest part in the paper.

Response:  The five phases was a framework to study the governance network.  The expectations were developed to create some learning points resulting from the research. In order to relate the expectations to the phases, two research questions were added in the introduction.  The Research Informed Expectations on Building Networks was moved to just before the results section and the expectations were tied to the research theorists. 

Analysis

4) This part should focus on the analysis. It should not introduce new concepts or arguments/findings from previous work. In many places, the authors do this, and hence it gives an impression that this section is mingled with the literature review. (e.g., discussions on Provan and Kenis (2007) in page 8 and Agranoff (2013)’s 10 network management lessons in page 10). I would just focus on the analysis based on the hypotheses proposed from the previous section.

            Response:  These are points well taken and the transcript has been modified to remove references to theory in the analysis.

Methods

5) This section should come first, not after the analysis. Currently, the authors use this section to justify the choice of ethnographic methodology. But what is missing is how the authors employed this ethnographic method. Plus, the authors’ justification of this methodology is far short of expectations. How do I know what the authors describe (regarding the program process) is correct? The authors argue that their descriptions are a result of years of in-depth observation and participation in the field (page 15). If so, the authors should provide a full description about how they got involved in observing (and participating in? what does this mean?) the program. The choice of ethnographic methods per se does not justify whether the authors’ observations and/or interpretation is valid. There should be some cross-references that can validate the authors’ findings. Or, at the very least, the authors are burdened to provide materials regarding how they tried to validate their interpretations. For example, “Data was drawn from governmental sources and policy directives” (p. 16). What data are you talking about? What governmental sources and policy directives were used? Also, what analytic approach the authors specifically (or strategically) used should be explained to augment the strength of ethnographic methods.

In sum, a more thorough method section that enhances credibility of the data sources and analysis needs to be provided rather a couple of self-justifying claims at the end.

Response:  The methods section was modified to include all government data obtained and utilized in the research. A further discussion was added to explain the connection of ethnographic methods. An entire section entitled Foreclosure Data was added.  Additionally integrative literature review was added as a methodological approach.  

Minor points

Background section look quite thin. I at least expected a couple of more sentences or one/two more paragraph about the background. It is very odd to see the next paragraph suddenly introduces the setup of the paper without the background having been sufficiently provided. The link to wicked problem is not well incorporated in the body. It was just used as a background. I would either delete it or incorporate this discussion somewhere in the analysis/discussion if the authors want to keep this part. “Cross-departmental collaboration” seems to be a better word than “network,” as this is essentially what the authors are examining. The term “phase” sounds awkward. Name it. Numerous grammatical errors and broken sentences degrade the quality of this manuscript. The authors should proofread the next iteration of this manuscript. For example, not “relative to”, but “related to.”

Response: These items have been considered and the manuscript amended to reflect these concerns. 

Round 2

Reviewer 1 Report

I appreciate the thoughtful consideration and application of recommended changes.

Reviewer 2 Report

The authors have adequately addressed the concerns and suggestions I have raised.